# Large area inkjet-printed OLED fabrication with solution-processed TADF ink

Chandra Kant[1,2], Atul Shukla [3,4], Sarah K. M. McGregor [3,5], Shih-Chun Lo [3,5] ✉, Ebinazar B. Namdas [3,4] ✉ & Monica Katiyar [1,2] ✉

This work demonstrates successful large area inkjet printing of a thermally activated delayed fluorescence (TADF) material as the emitting layer of organic light-emitting diodes (OLEDs). TADF materials enable efficient light emission without relying on heavy metals such as platinum or iridium. However, low-cost manufacturing of large-scale TADF OLEDs has been restricted due to their incompatibility with solution processing techniques. In this study, we develop ink formulation for a TADF material and show successful ink jet printing of intricate patterns over a large area (6400 mm²) without the use of any lithography. The stable ink is successfully achieved using a non-chlorinated binary solvent mixture for a solution processable TADF material, 3-(9,9-dimethyla-cridin-10(9H)-yl)-9H-xanthen-9-one dispersed in 4,4'-bis-(N-carbazolyl)-1,1'-biphenyl host. Using this ink, large area ink jet printed OLEDs with performance comparable to the control spin coated OLEDs are successfully achieved. In this work, we also show the impact of ink viscosity, density, and surface tension on the droplet formation and film quality as well as its potential for large-area roll-to-roll printing on a flexible substrate. The results represent a major step towards the use of TADF materials for large-area OLEDs without employing any lithography.

Due to their attractive features such as high brightness, fast response time, wide viewing angle, low power consumption, and high flexibility, organic light-emitting diodes (OLEDs) have been at the forefront of next-generation flat-panel displays and solid-state lighting technologies[1–5]. Unfortunately, current OLED manufacturing requires sophisticated vacuum deposition techniques, which are low-yielding, and energy and material intensive. Therefore, it is highly desirable to replace this production method with solution-based approaches that can be fabricated under fast and simple ambient conditions at lower cost[6–9]. Inkjet printing (IJP) is the most promising technology for solution deposition of high-resolution patterns for high-quality OLED displays[10]. IJP offers the primary advantage of drop-on-demand, where precise drop placement of the ink can be automated and modified in-line during device fabrication to create customisable and adjustable patterns without the need of masks[11–14]. This provides tremendous capability in developing prototypes as well as adjusting small-scale to medium-scale production. Furthermore, in contrast to thermal/vacuum deposition where only a small fraction of the materials reach the substrate, printing with an inkjet machine can utilize almost all of the ink, which is both economically favourable and environmentally friendly[15–18].

The IJP technique is gaining favour for large-area high-resolution patterning due to its precise drop placement and mask-free deposition[19–23]. However, to employ IJP for large-scale commercial OLED display applications, certain prerequisites must be fulfilled, including: (i) solution processable low-cost triplet emitters; (ii) printing from non-chlorinated green solvents; (iii) large-area printing in ambient conditions without compromising the device performance; and (iv) high-resolution patterning without the use of traditional

[1]Materials Science and Engineering Department, Indian Institute of Technology Kanpur, Kanpur, India. [2]National Centre for Flexible Electronics, Indian Institute of Technology Kanpur, Kanpur, India. [3]Centre for Organic Photonics & Electronics, The University of Queensland, Brisbane, Australia. [4]School of Mathematics and Physics, The University of Queensland, Brisbane, Australia. [5]School of Chemistry and Molecular Biosciences, The University of Queensland, Brisbane, Australia. ✉e-mail: s.lo@uq.edu.au; e.namdas@uq.edu.au; mk@iitk.ac.in

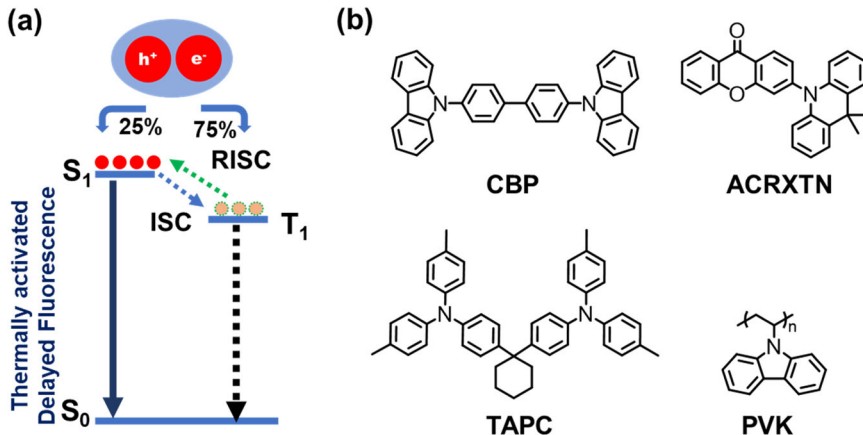

**Fig. 1 | Working principle of TADF and chemical structures of materials used in the study. a** Simplified Jablonski diagram of TADF route, showing electron-hole recombination. ISC indicates the intersystem crossing, RISC is the reverse intersystem crossing, $S_0$ is the ground state, $S_1$ is the first singlet excited state, and $T_1$ is the first triplet excited state. **b** Molecular structures of CBP, ACRXTN, 1,1-bis[(di-4-tolylamino)phenyl]cyclohexane (TAPC), and polyvinylcarbazole (PVK).

lithography. This requires the emissive materials to be formulated into a stable ink with a suitable viscosity and surface tension range[24–26], which has been a challenging task. The ideal ink should also provide a smooth, uniform coating without thickness variation[27–31]. Hence, low-cost ink formulation must be developed for uniform jetting and optimal film formation for roll-to-roll printing.

Thermally activated delayed fluorescence (TADF) emitters have gained significant interest in recent years because of their ability to harvest triplet excitons for efficient electroluminescence (EL) without the need of expensive and rare metals like iridium or platinum[32–39]. TADF emitters allow for efficient triplet to singlet spin-flip due to their narrow singlet-triplet energy gap ($\Delta E_{S-T}$). The small $\Delta E_{S-T}$ allows reverse intersystem crossing (RISC) to effectively take place at room temperature[40,41], leading to repopulation of singlet excited states for light emission (Fig. 1). In 2012, Adachi and co-workers demonstrated that the TADF mechanism could be critical in developing a next generation of highly efficient OLEDs[42–45]. Most of the reported TADF materials, however, require high temperature and high vacuum evaporation deposition. Solution processable TADF materials compatible with IJP remains rare[24–26,46]. This is mainly due to challenges in the development of a stable high-performance TADF ink with appropriate viscosity and surface tension, without the use of any chlorinated solvents.

In this work, we demonstrated high-performance TADF ink formulation for OLED fabrication without employing halogenated solvents. The ink formulation comprises of 3-(9,9-dimethylacridin-10(9H)-yl)-9H-xanthen-9-one (ACRXTN) dispersed in 4,4′-bis-(N-carbazolyl)-1,1′-biphenyl (CBP) host matrix, using a non-chlorinated binary solvent mixture of toluene and methyl benzoate (MB). We also provided a systematic study of the variables that affect IJP for material deposition, including droplet formation, wetting properties, IJP film quality and solidification, as well as its potential for large-area roll-to-roll printed OLEDs. First, we examined the impact of ink viscosity, density, and surface tension on droplet formation. Then, we achieved large-area IJP pattern on a flexible polyethylene terephthalate (PET) substate (80 × 80 mm²) at room temperature and ambient conditions. Finally, we demonstrated large-area IJP OLEDs with performance comparable to the control spin-coated OLEDs.

## Results

### TADF material synthesis

TADF material, ACRXTN, was synthesized using 9,9-dimethyl-9,10-dihydroacridine and 3-bromo-9H-xanthen-9-one under Buchwald cross-coupling by modification to reported procedure[47]. Details of synthesis can be found in Supplementary Note 1.

### Ink formulation of the emission layer

To create a homogeneous and smooth emissive thin film, the first step was to select suitable solvents and concentrations for the emissive blend containing ACRXTN TADF guest and CBP host. The ink formulation using combination of non-chlorinated solvents with a low and a high boiling point was tried[48,49] where more details with other solvents are given in the Supplementary Note 2 and Table S1. We found that while low boiling point solvents evaporate rapidly at the air–nozzle interface, a higher boiling point solvent (e.g., o-DCB) solves the evaporation issue but it restricts the drying mechanism of drops after jetting, and chlorinated solvents lead to poor long term ink stability. Hence, we used non-chlorinated solvents of toluene and methyl benzoate (MB) as a binary solvent system in this work. We noted that MB has a relatively higher viscosity (2.05 cP), higher boiling point, and low vapour pressure to act as the principal solvent. However, it has a comparatively high surface tension of 37.5 mN m⁻¹, which is slightly over the optimal range for IJP. To overcome this, toluene was selected as the co-solvent because of its lower boiling point (110 °C) and a lower surface tension (28 mN m⁻¹), allowing for the final surface tension to be adjusted to the optimal range (28–32 mN m⁻¹) of the Fujifilm Dimatix Samba Cartridges (DMC-11610) used in this study. The ink formulation was found to have a high stability exceeding one week using this recipe.

Next, we experimentally checked the stability of the ink with various concentrations, including 5.5, 11.25, 15, and 20 mg mL⁻¹ of TADF/CBP in toluene:MB (40:60) and found that the 5.5 and 11.25 mg mL⁻¹ ink formulation gave a stable ink (with more detailed results tabulated in Supplementary Note 2 and Table S2). Even though the 5.5 mg mL⁻¹ ink formulation was quite stable for more than a month, inkjet printing with this formulation produced very thin films (≈10 nm) (Supplementary Figure S4). In order to increase thickness of the films, we increased the concentration. However, we found that while the concentration >15 mg mL⁻¹ did not result in stable ink at ambient temperature. Therefore, we employed our ink formulation with 11.25 mg mL⁻¹ for the inkjet printing study.

Table S3 presents the physical characteristics of the ink with a concentration of 11.25 mg mL⁻¹ and formulation dissolved in a combination of toluene:MB (40:60). The Z number is often utilized to determine printability and characterize droplets in IJP (For more detail refer the Supplementary Note 3)[50]. Droplets having a Z number between 1 and 10 should theoretically print correctly without indicating the existence of satellites or tails during emission[51], while ink with a Z value greater than 14 has been documented to exhibit constant jetting behaviours without breaking of uniform droplets[52]. Even though

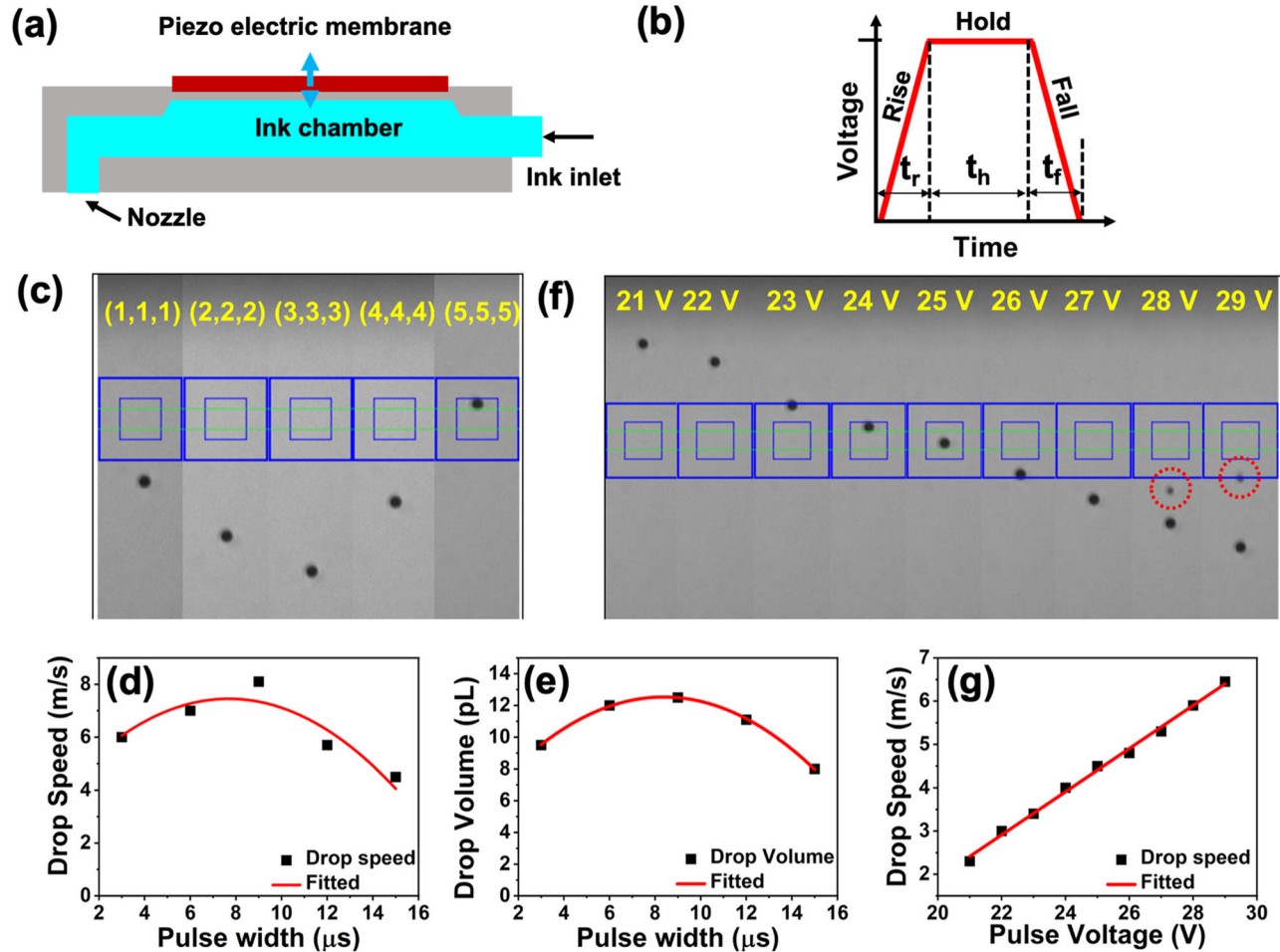

**Fig. 2 | Optimising the ink jet printing parameters. a** Schematic diagram of the cartridge printhead. **b** Standard unipolar waveform used to drive the piezo-printhead having three major segments rise time, hold time, and fall time; showing the mechanism of droplet formation. **c** Images captured with increasing pulse width shows that at 9 μs (3,3,3) pulse has the highest drop speed. **d** Nonlinear plot of drop speed variation with change in pulse width from 3, 6, 9, 12, and 16 μs, respectively. **e** Drop volume variation with change in pulse width. **f** Effect of pulse voltage on the drop velocity with applied pulse width of 9 μs ($t_r = 3$, $t_h = 3$, and $t_f = 3$). Drop images captured with an inbuilt drop watcher camera with increasing pulse voltage (12 to 29 V) at pulse width of 9 μs ($t_r = 3$, $t_h = 3$, and $t_f = 3$). **g** A linear relationship between the applied pulse voltage with the drop speed.

our developed ink has a Z value of 12.9, we were still able to achieve consistent jetting conditions by adjusting the waveform that drives the printhead.

### Effect of pulse width on drop velocity and volume

Figure 2a depicts a simplified version of a printhead consisting of a piezoelectric membrane that flexes in response to a changing voltage waveform, allowing the ink to fill and drain. Unipolar pulse waveforms were used, which may be broken into three major parts defined by ($t_r$, $t_h$, $t_f$)[53,54] where, $t_r$, $t_h$, $t_f$ are the rise time, hold time and fall time, respectively. As shown in Fig. 2b, the piezoelectric membrane is pulled by the rising time after the ink has entered the nozzle from the reservoir, and pushed by the falling time while ink droplets are ejected from the nozzle. This is because the ink chamber experiences negative pressure during the pulling stage and a positive pressure during the pushing stage. Even after the initial droplet has been ejected from a piezoelectric inkjet printer, the actuating membrane vibrates back and forth, causing residual vibrations in the ink channel that may reduce print quality[51,55–60]. Since the jetting frequency plays a crucial role in our tests, we held it constant at 1 kHz. Drop watcher was used to observe the drops in real-time, displaying the value of drop speed and volume with firing angle.

The influence of pulse width on the generated droplet is shown in Fig. 2c, where we kept the applied voltage constant at 27 V and tested different pulse widths of 3 (1,1,1), 6 (2,2,2), 9 (3,3,3), 12 (4,4,4), and 15 (5,5,5) μs. We observed that in accordance with the pulse width of the driving waveform, the droplet velocity and volume changed as shown in Fig. 2d, e.

The velocity of the droplets was found to rise as a function of the driving voltage. As shown in Fig. 2f, the droplet produced with higher voltages (>28 V) have "satellites" or "tails" shown with red dotted circles. We observed no drops were produced at voltages below 17 V. The rate at which drops are produced can be adjusted by varying the voltage delivered to the piezoelectric element in the cartridge settings box. A linear relationship between the droplet speed and pulse voltage was observed (Fig. 2g). We also discovered that reducing the number of firing nozzles from all (16) to few (5 or 1) reduced the drop velocity. The maximum drop speed was 8 m s$^{-1}$ with a drop volume of 12.3 pL at the 9 (3,3,3) μs pulse width.

### IJP on PEDOT:PSS films

After improving the jetting parameters, the ink was printed on the hole injection layer (PEDOT:PSS) as shown in Figure S5. However, the ink showed significant dewetting effects while drying. In order to improve

the wetting characteristic of the TADF ink, a PVK-TAPC blend thin film (15 nm) was spin-coated on the PEDOT:PSS layer. The layer also serves as an energy barrier bridge between the emissive layer and the hole injection layer. The TAPC-PVK blend film significantly improved the wettability of the TADF film. Furthermore, hole transport between the PEDOT and TADF layers is bridged by a layer of PVK-TAPC blend, which also serves as an electron-blocking layer[61,62]. They work well together to produce a high-quality film using a chlorobenzene solvent, and the resulted film has improved wetting for the TADF ink.

Drops were printed at 200 dots per inch (DPI) at three different substrate temperatures of 30, 40, and 50 °C. The drying behaviours of the printed patterns including drop diameter, ink segregation, droplet shape upon impinging on the substrate, and deposit thickness profile were investigated using optical profilometer (Fig. 3). At a lower substrate temperature of 30 °C, individual droplets had a diameter of $110 \pm 2 \, \mu m$, and their M-shaped structure showed material build-up at the edges in a ring formation (Fig. 3a1–d1)[63]. This caused TADF ink to move in order to compensate for liquid lost due to evaporation at the contact line. The final deposit had a distinct ring at the perimeter to the solidification at the contact line. As we increased the substrate temperature to 40 °C, the droplet shape and the diameter were altered. Instead of an M-shape, it appeared a bell shape structure with a peak height of 50 nm with a drop diameter of $80 \pm 2 \, \mu m$, and the coffee ring effect completely vanished at this temperature (Fig. 3a2–d2). The Marangoni effect may be activated in conjunction with a solvent combination (with varying vapour pressures and boiling

temperatures) to counteract the coffee ring effect[64,65]. Processing the substrate at 40 °C can drastically reduce the coffee ring effect. As shown in Fig. 3a3–d3, upon increasing the substrate temperature to 50 °C, the shape and diameter of the droplets changed from a distinct bell shape to the appearance of a mountain hill with a plateau structure having an average drop diameter of $75 \pm 2 \, \mu m$ with a peak height of 35 nm. At this temperature, solvents started to simultaneously evaporate with droplets imping on the substrates and the contact line was pinned faster than printed at the lower substrate temperature, resulting in a smaller drop diameter. Hence, the substrate temperature is critical in controlling the shape and the diameter of inkjet-printed droplets since it occurs during the drying of liquid beads and results in a unique dual-ridged line profile after solidification.

## Print resolution on the film formation process

To learn more about the change from discrete drops to a continuous layer film, we examined the influence of DPI in resolution range from 100 to 800 DPI at 40 °C substrate temperature. Figures 4 and S6 show the optical images of printed TADF droplets at different DPIs @ 40 °C substrate temperature. The TADF droplet below 400 DPI on ITO/PEDOT:PSS/PVK-TAPC exhibited discontinuities. Drops were separated by large distances preventing them from interacting with each other as they spread. When the DPI was raised, the gaps between the individual droplets were filled in, creating a seamless overlay. As shown in Fig. 4c–e, as DPI was increased over 400, droplets began to overlap and combined to form a continuous film. Hence, we can conclude that

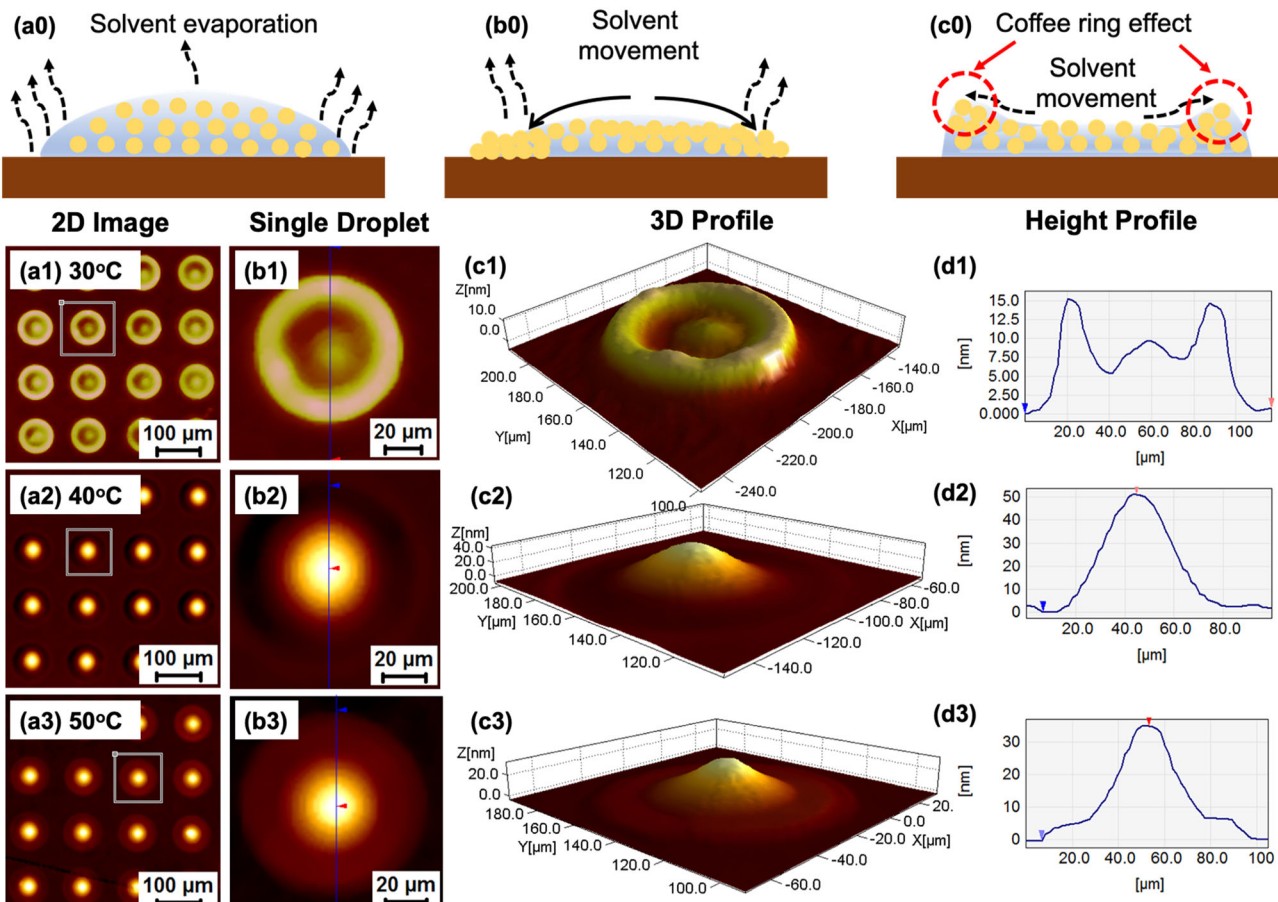

**Fig. 3 | Coffee ring formation schemes, 2D & 3D profiles, and height profiles. a0, b0, c0** Schematic diagram for the coffee ring formation. **a1–3** Shows the 2D-images of inkjet printed patterns (taken with an optical profilometer) on PVK-TAPC films with 200 DPI with three substrate temperatures at 30, 40 and 50 °C, respectively. **b1–3** 2D-images of the single droplet at 30, 40 and 50 °C substrate temperature,

respectively. **c1–3** Drop's 3D profile exhibits an M-shape, a bell shape and a mountain shape with a plateau at lower foot, respectively. **d1–3** Height profile of the single drop.

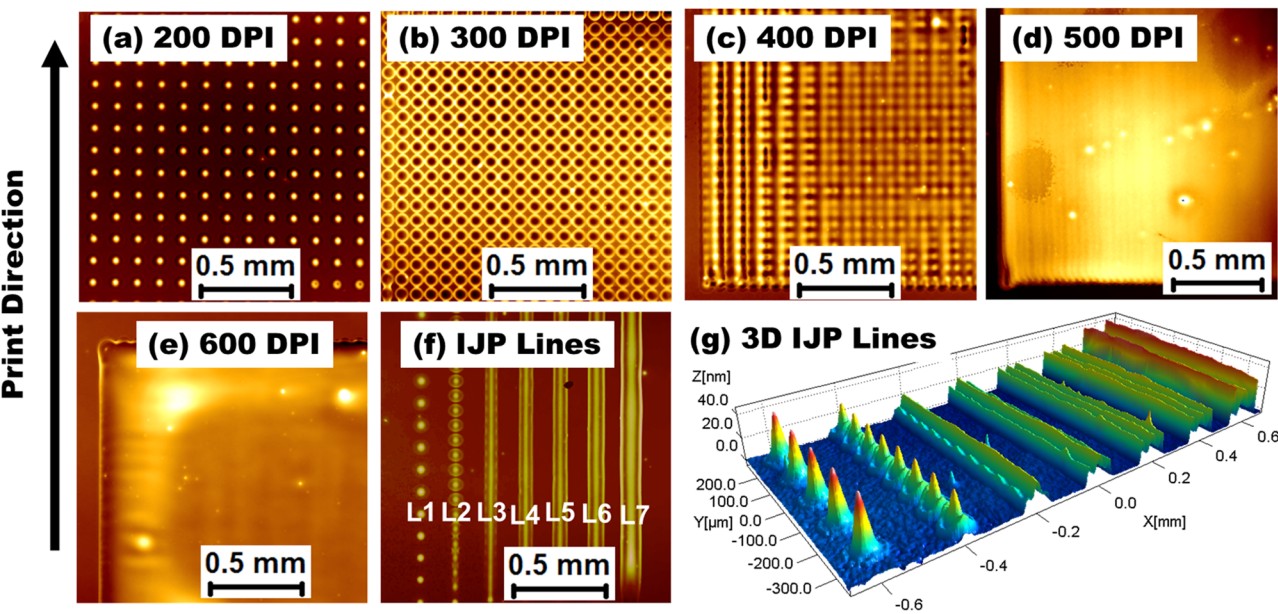

**Fig. 4 | The microscopic image of printed droplets in 4 × 4 mm² square patterns with different DPIs on substrates @ 40 °C. a** Printed with 200 DPI. **b** 300 DPI. **c** 400 DPI. **d** 500 DPI. **e** 600 DPI. **f** Printed lines with 200 to 800 DPI from L1 to L7, respectively. **g** 3D-images of IJP single lines L1 to L7 (200 to 800 DPI).

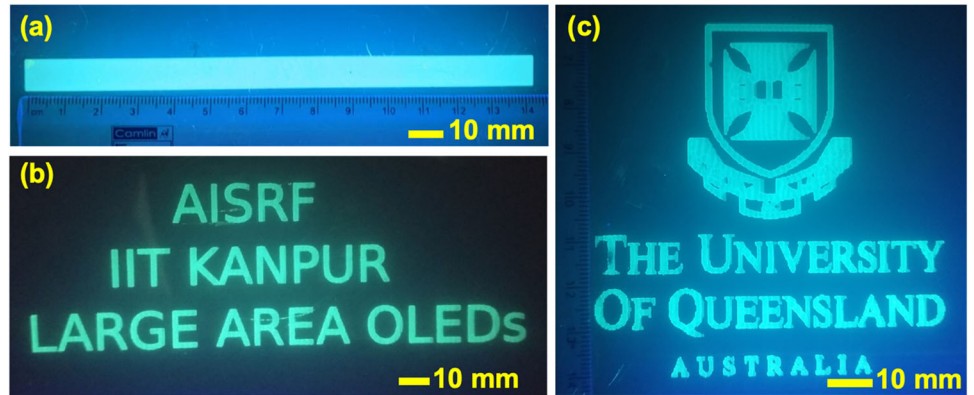

**Fig. 5 | Inkjet-printed large area strip, texts and a figure image under UV illumination. a** Inkjet-printed large rectangular strip (10 × 145 mm²) on a flexible PET substrate. **b** IJP large area texts of "*AISRF IIT KANPUR LARGE AREA OLEDs*" on 50 × 150 mm² PET substrate. **c** IJP *The University of Queensland* logo on a PET substrate, all patterns are printed with 600 DPI resolution (with all patterns being illuminated with 325 nm UV light).

resolution over 400 DPI are necessary to generate consistent films at 40 °C substrate for our formulated ink. Figure 4f, g shows the printed single lines of TADF with increasing DPIs from 200 to 800. As the DPI increased, the linewidth broadening also increased from L1 to L7 at 75.25, 75.25, 76.10, 77.99, 84.41, 97.45, and 126.62 μm, respectively (Figure S7). At higher DPI of 600, coffee ringing was observed, suggesting that the co-solvent content was under-optimized. We learn from printed patterns that DPI may be used to precisely govern characteristics of the smallest feature size analogous to individual ink droplets. When comparing the linewidth at 400 and 800 DPI, there was around 40% increase for 800 DPI (126.62 μm) over that (76.10 μm) of 400 DPI. The primary reason for this is that when the DPI rises, more droplets are poured into the same region, making the pattern thicker and broader. Hence, it is important to consider the exact DPI for the desired thickness. Finally, by manipulating these printing settings, one may get covered surfaces, lines, or well-defined spots by controlling the drop space by adjusting the DPI, to allow functional material deposition with no extra processes. Atomic microscope images of these IJP films were smooth with the films' RMS roughness (R_q)

values < 1 nm (see Figure S8). This finding reveals that IJP of TADF ink can create uniform thin film by fine-tuning the printing parameters. This uniformity is necessary for multilayer light-emitting devices, demonstrating why this finding is significant.

## IJP of a strip, texts and a figure image on large-area PET flexible substrates

To show the adaptability and potential of this ink for large-area OLEDs, IJP of various creative forms and patterns were studied. IJP enables the possibility to move away from needing to use expensive and time-consuming lithographic processes when producing signs and/or complex patterns. Furthermore, IJP for roll-to-roll processing of TADF ink allows for maskless design, making it an ideal approach to launch mass production. Herein, we demonstrate the feasibility of large area IJP at standard room temperatures and humidity levels after waveform optimization with a small-scale lab printer utilizing solely non-halogenated solvents. Figure 5 shows 600 DPI TADF inkjet-printed large rectangular strip (10 × 145 mm²) and large area texts of "*AISRF IIT KANPUR LARGE AREA OLEDs*" on 50 × 150 mm² and, *The University of*

*Queensland* logo on a flexible PET substrates (all patterns are illuminated with a 325 nm UV lamp). The additional printed images are shown in Supplementary information in Figure S9.

## IJP small area non-patterned OLEDs (with an active area of 4 × 4 mm²)

To study electroluminescence properties of the TADF ink, small area OLEDs were fabricated on ITO-coated glass substrates. Device fabrication steps are shown in Fig. 6. After performing an ozone surface treatment on ITO substrate, we sequentially deposited PEDOT:PSS (40 nm) and PVK-TAPC blend (15 nm) using spin-coating technique. Films were annealed at 120 °C for 15 min. The TADF ink was ink jet printed on PVK-TAPC blend film using an LP50-Pixdro industrial-grade inkjet printer equipped with a piezoelectric-driven 10 pL Dimatix cartridge (16 nozzles of 21 μm diameter). For direct comparison, the same ink formulation was spin-coated on PVK-TAPC blend film. Both inkjet-printed and spin-coated TADF films were annealed at 90 °C for 10 min under a nitrogen shower. The substrates were then moved in a vacuum chamber, and TPBi (32 nm), calcium (20 nm), and aluminium (100 nm) were sequentially evaporated by thermal evaporation at $5 \times 10^{-6}$ torr.

Inkjet-printed OLEDs (IJP1, IJP2, and IJP3) with varying thicknesses were achieved by changing the resolution from 500, 600, and 700 DPI, respectively, while device SP1 was the spin-coated TADF OLED. The solvent composition for the inkjet-printed and spin-coated devices were the same, and all the devices had an identical device architecture of ITO/PEDOT:PSS (40 nm)/PVK-TAPC (15 nm)/TADF/TPBi (32 nm)/Ca (20 nm)/Al (100 nm) with an active area of 4 × 4 mm².

Figure 7 shows current-voltage-luminescence characteristics of the OLEDs. The current density in IJP1 peaked at 100 mA cm⁻², while the current density in the SP1 (with a thicker active layer than that of IJP1) reached only 41 mA cm⁻² at the same voltage. As shown in Fig. 7b, the turn-on voltages ($V_{on}$) corresponding to 1 cd m⁻² were found to be 4.0, 4.1, 4.2, and 4.3 V for SP1, IJP1, IJP2, and IJP3, respectively. The IJP and spin-coated OLEDs showed comparable brightness level exceeding 2000 cd m⁻². As shown in Fig. 7c, SP1 had a maximum current efficiency of 13.7 cd A⁻¹ @ 100 cd m⁻². In contrast, the maximum current efficiencies of inkjet-printed devices IJP1 (500 DPI), IJP2 (600 DPI) & IJP3 (700 DPI) were 12.4, 7.5, and 9.5 cd A⁻¹ @ 100 cd m⁻², respectively. The highest current efficiency corresponded to the 500 DPI device (IJP1), which has a thinner TADF layer, resulting in a high current density and maximum

luminescence. We note that as the thickness increases, the light turn-on voltage, and the device efficiency decreases. Table 1 summarizes device characteristics of 3 to 4 pixels per each device. At 1000 cd m⁻², pixel to pixel device variation is less (~5%), indicating uniform and reproducibility of IJP films. The maximum external quantum efficiency of the devices (EQE) were ~4.7% @ 30 cd m⁻² for spin-coated and ~4.3% @ 30 cd m⁻² for the IJP OLEDs (Figs. S10, 11). It is interesting to note that the efficiency roll-off of the IJP devices were comparable to the spin-coated device. At a higher brightness level, the efficiency of IJP OLEDs and spin coated OLEDs were ≈7 ± 1 cd A⁻¹ @ 1000 cd m⁻². This suggests, charge balance is maintained in simple device structure (i.e., without blocking layers) and exciton annihilation is negligible in the emissive layer. The EL spectra of the spin coated (SP1) and IJP1 (500 DPI), IJP2 (600 DPI) & IJP3 (700 DPI) at 1000 cd m⁻² were almost identical (Fig. 7d). This suggest, exciton formation and recombination photo physics of inkjet-printed OLEDs were identical to the spin coated OLEDs.

## IJP small area patterned OLEDs (with active areas of 4 × 4 mm² and 37 × 37 mm²)

Patterning is essential if an OLED is to be used for anything other than a simple light source, such as for signage or logo applications. The electrode may be defined in several ways, including lithographically patterning an ITO electrode, depositing a dielectric layer on the substrate, or deposition via a shadow mask. Photolithography is a multi-stage process that includes covering with a photoresist, exposing to ultraviolet light, and etching ITO. Ink jet printing approach does not include the patterning of electrodes but rather the selective deposition of an active layer over the entire substrate. Hence, it does decrease the number of steps required to manufacture a device. For this reason, the presence of the active TADF layer results in the selective emission of light. This method allows in-situ patterning to be performed without requiring further fabrication steps, eliminating the need for electrode patterning.

Figure 8b shows the TADF OLEDs with an active area of 37 × 37 mm², in which the emission region was directly printed in the form of the text "*AISRF OLED*" on an un-patterned ITO substrate without any masks. A 600 DPI IJP devices shows smooth emission and has sharp borders (Inset image Fig. 8). In addition, we also observed a faint brownish-pink background hue (15–20 cd m⁻²) as the voltage increases, which was resulted from direct charge carrier recombination from TPBi to PVK-TAPC blend film. Similarly, in the literature, Wang et al.

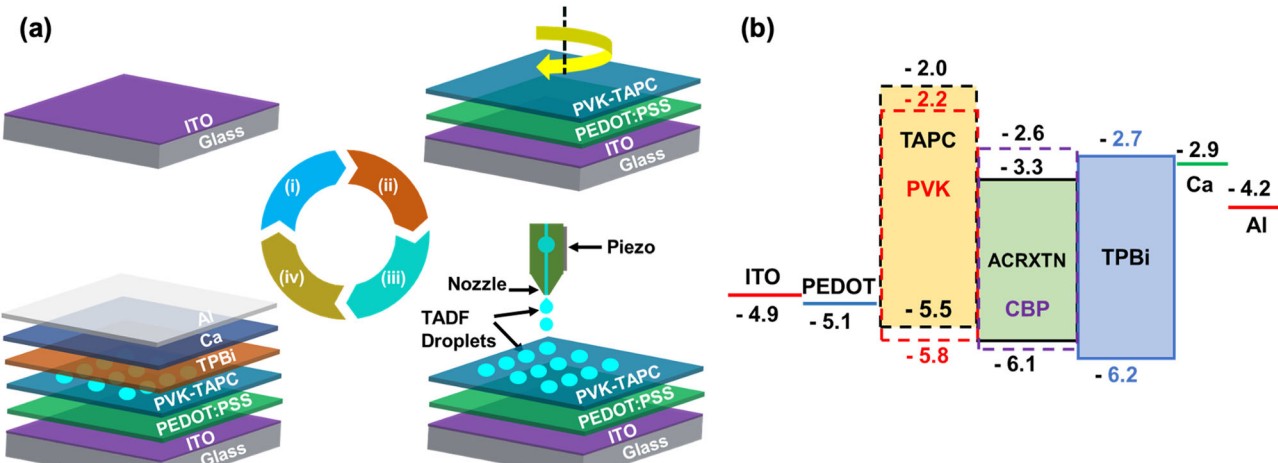

**Fig. 6 | Device fabrication processes and energy level diagram for the inkjet-printed OLEDs. a** Device fabrication steps (i) ITO-coated glass substrate; (ii) spin-coated PEDOT:PSS and PVK-TAPC blend layer on ITO; (iii) IJP of TADF ink on PVK- TAPC; (iv) thermal deposition of other functional layers: TPBi (32 nm), Ca (20 nm) and aluminium (100 nm). **b** Energy level diagram of the OLED layers (all values are in eV).

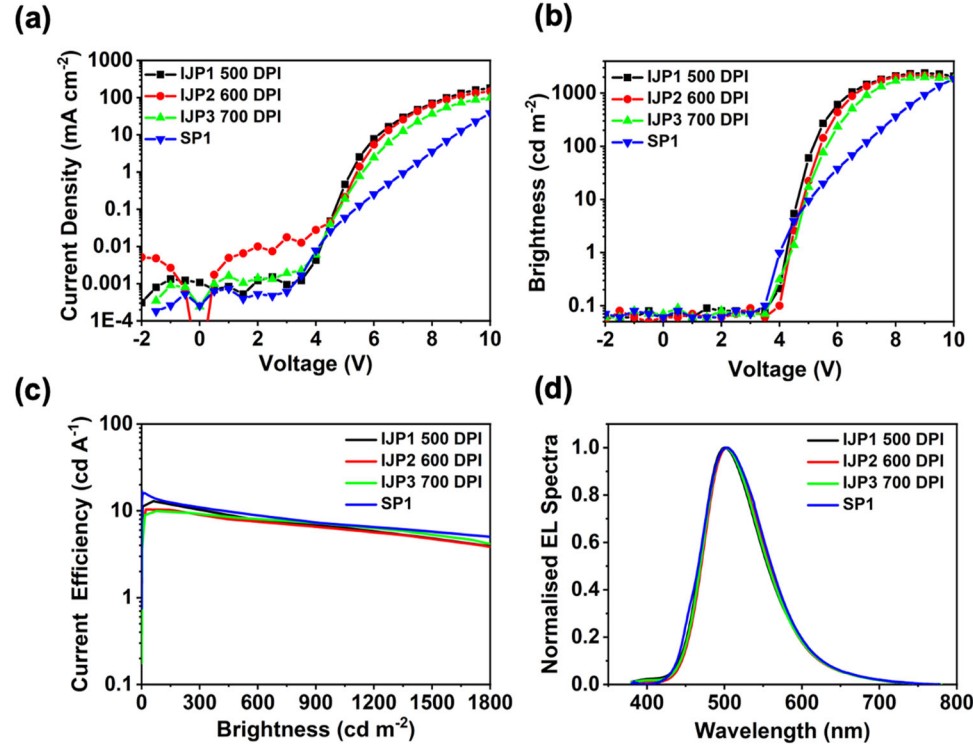

**Fig. 7 | Device characteristics and EL spectra of the IJP 1-3 and spin-coated (SP1) OLEDs. a** Current density *versus* voltage in log-linear scale. **b** Luminance *versus* voltage. **c** Current efficiency *versus* brightness characteristics of the devices. **d** EL spectra of spin coated (SP1) and inkjet-printed devices recorded at 1000 cd m⁻².

**Table 1 | Summary of $V_{on}$, $LE_{max}$, $L$, and CIE coordinates of the fabricated devices**

| Devices | TADF thickness (nm) | $V_{on}$ (V) | CE (cd A⁻¹) @ 100 cd m⁻² | CE (cd A⁻¹) @ 1000 cd m⁻² | $L_{max}$ (cd m⁻²) |
|---|---|---|---|---|---|
| SP1 | 36 ± 2 | 3.9 ± 0.1 | 13.7 ± 0.8 | 7.6 ± 0.8 | 1775 ± 80 |
| IJP1 (DPI = 500) | 25 ± 2 | 4.1 ± 0.1 | 12.4 ± 1.0 | 6.8 ± 0.5 | 2350 ± 40 |
| IJP2 (DPI = 600) | 29 ± 2 | 4.2 ± 0.1 | 7.5 ± 2.7 | 6.0 ± 0.2 | 2060 ± 90 |
| IJP3 (DPI = 700) | 33 ± 3 | 4.3 ± 0.1 | 9.5 ± 0.4 | 6.9 ± 0.6 | 1985 ± 10 |

Device pixel area 4 × 4 mm².

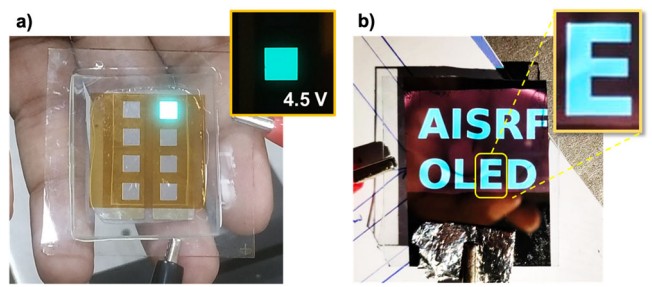

**Fig. 8 | Images of small area OLEDs. a** An operational inkjet-printed TADF OLED biased at 6.0 V emission of cyan colour light with an active area of 4 × 4 mm², and an inset image of the same pixel at a lower voltage bias (4.5 V). **b** An ink jet patterned OLED with texts of "*AISRF OLED*" on 37 × 37 mm² area in a room light (biased at 5.0 V), whereas the inset image of letter 'E' shows the sharp edges of printed patterns.

have shown that energy transfer can occur between TPBi to NPB host in their phosphorescent OLEDs through the interfacial exciples host (IEH)[66]. This could be the possible reason for the weak light emission where TADF ink was not printed.

## IJP large-area non-patterned OLEDs (with an active area of 80 × 80 mm²)

To demonstrate large area ink-jet printed (LAiP) OLEDs, we fabricated large-area panel (LAP) IJP TADF OLEDs on 120 × 120 mm² chromium/ITO/glass substrates with an active area of 80 × 80 mm² (Fig. 9a) in a 10,000-class clean room at ambient conditions. We applied the chromium gridlines for the homogeneous current distribution as the sheet resistance of ITO starts to affect the large-area substrates. After performing an ozone surface treatment of the ITO substrate, PEDOT:PSS and PVK-TAPC layers were spin-coated on top. Substrates were then annealed at 120 °C for 15 min before the TADF ink was IJP at 600 DPI on top of the PVK-TAPC layer. This gave a device structure of ITO/PEDOT:PSS (40 nm)/PVK-TAPC (18 nm)/TADF (30 nm)/TPBi (32 nm)/Ca (20 nm)/Al (100 nm). The devices were then encapsulated for characterization and analysis.

Figure 9c shows an operational LAiP TADF OLED at 10.5 V. The device emitted homogenous light with a maximum luminescence of 622 cd m⁻². We noticed some non-uniformities of light emission at the lower voltages but they disappeared as the bias voltage increased. This may be due to non-uniformity of the spin-coated layers (PEDOT:PSS and PVK-TAPC) on a large substrate area. We also observed that the presence of dust particles on the substrate during the annealing process, which resulted in devices getting shorted. To minimize dust

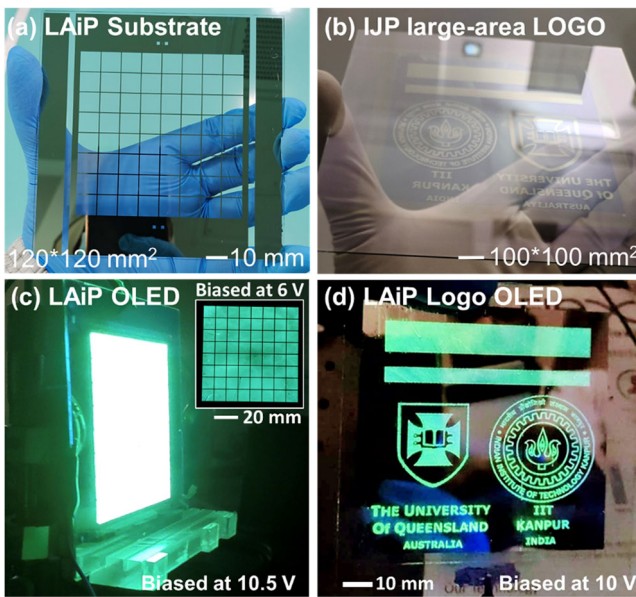

**Fig. 9 | Images of large-area ink jet panel substrates and OLEDs. a** Handheld large-area panel (LAiP) substrate (120 × 120 mm²) with an active area of 80 × 80 mm² through square chromium gridlines. **b** Inkjet patterned large area logos on a 100 × 100 mm² substrate, consisting of *The University of Queensland* and *IIT Kanpur* with two rectangular strips of 10 × 70 and 5 × 70 mm² before the thermal evaporation of other layers. **c** The LAiP OLED (120 × 120 mm²) operated at 10.5 V with an inset image at 6 V. **d** The LAiP (100 × 100 mm²) intricate logo OLED biased at 10 V with brightness of ≈500 cd m⁻². Brightness measured at the rectangular strip.

particles, substrates annealing was, therefore, performed under N₂ shower. With few iterations, we successfully fabricated and tested the large-area devices. As shown in Figs. 9c and S12, the LAiP OLED had the maximum current efficiency of 13.7 cd A⁻¹ @ 622 cd m⁻², which is close to those of spin-coated small area OLEDs.

### IJP large-area high-contrast patterned OLEDs (with an active area of 80 × 80 mm²)

Next, we demonstrate maskless IJP patterning technique to create intricate OLEDs on a large area substrate. The device fabrication methods are outlined in the Supplementary Figure S13. First, we inkjet printed a SU-8 (a negative epoxy) based photoresist dielectric (DPI 800; thickness 1 μm) to create the desired intricate template on 100 × 100 mm² ITO substrates. The resulted SU-8 patterned layer was then exposed with UV light to cross-link the photoresist. HIL (PEDOT:PSS) and HTL layers (PVK-TAPC blend) were then spin coated on top of the SU-8/ITO template. The TADF ink was IJP at 600 DPI on the PVK-TAPC layer to create the intricate light emission regions, matching with the dielectric template. Finally, the substrates were transferred to an evaporator for thermal deposition of TPBi (32 nm) and Ca/Al electrodes. Figure 9d shows the proof-of-concept device consisting of logos of The University of Queensland and IIT Kanpur as well as two rectangular strips of 10 × 70 and 5 × 70 mm². The device achieved brightness of 500 cd m⁻² at 10 V. The images show high contrast glowing edges revealing intricate features. It is important to note that our negatively IJP patterned SU-8 dielectric layer eliminates direct charge recombination from TPBi to PVK-TAPC regions. This leads to OLEDs with high contrast images without the use of complicated photolithography.

### Discussion

We have developed large-area multilayer OLEDs with an inkjet-printed TADF emission layer. We have successfully demonstrated IJP as a single step maskless emission layer patterning technique, to create intricate and high-resolution designs for signage, wearable electronics, and advertising without the use of typical lithography steps. We achieved stable ink formulation using non-chlorinated binary solvent mixture of toluene:MB in combination with a suitable quantity of TADF and CBP. We emphasise that the formulated ink with the mixed solvent system has no adverse effects on the existing organic layers. Printed films with a resolution of 500 DPI or greater had nanoscale roughness of less than 1 nm. The best-printed OLEDs have a maximum brightness of 2389 cd m⁻² and a maximum current efficiency of 12.4 cd A⁻¹ @ 100 cd m⁻². In contrast, the spin-coated devices have a maximum brightness of 1873 cd m⁻² and a maximum current efficiency of 13.7 cd A⁻¹ @ 500 cd m⁻². Since a computer processes the pattern, any forms may be designed and printed quickly, enhancing the manufacturing speed and unlocking a new frontier for the printed electronic sector. These findings pave the door to a new generation of TADF emissive materials for use in roll-to-roll inkjet-printed OLEDs on large areas.

## Methods

### Material synthesis

TADF material, ACRXTN, was synthesised with modification to a reported procedure[47]. Reaction scheme, synthetic procedure, and characterisation data can be found in the Supplementary Note 1.

### OLED Device fabrication process

First, ITO-coated glass substrates with a sheet resistance of 15 Ω sq⁻¹ were ultra-sonicated in deionized water, acetone, and isopropyl alcohol for 15 min each, followed by 15 min of drying on a hotplate at 120 °C for 10 min. The wettability of the ITO substrates were then increased by exposing them to UV ozone for 12 min. PEDOT:PSS solution was spin coated on the ITO substrate at 4000 RPM for 60 seconds and annealed at 120 °C for 15 min in the air. The PVK-TAPC blend film was prepared by mixing PVK:TAPC (3:1 wt%) in chlorobenzene (5 mg/ml) solution. The solution was spin-coated on ITO/PEDOT:PSS substrate and annealed at 120 °C for 15 min to give a ≈ 15 nm thick film. The TADF ink was spin coated at 2000 RPM and annealed at 90 °C for 10 min in the N₂ shower. For inkjet printing, a 10 pL Dimatix cartridge with a nozzle diameter of 21 μm was used. A square area of 4 × 4 mm² was dedicated and defined to producing visible light for both inkjet-printed and spin-coated devices. A UV-curable epoxy droplet was placed on top of an active pixel and gently squeezed with a glass layer to cure the epoxy. A UV lamp was used to expose the sample to UV light for encapsulation of final device.

### Ink formulation and characterization

The TADF ink was prepared by mixing 12.5 wt% of ACRXTN in CBP host in solvent mixture of MB and toluene (40:60) with total concentration of 11.25 mg mL⁻¹ for the final ink. Before using the solution, it was agitated for 2 h at room temperature. Ink viscosity has been measured with rolling-ball viscometer (Anton Paar -Lovis 2000 M/ME) and the surface tension values are measured with Goniometer (OCA 15EC).

### Ink and device characterization

The thickness and 3D profile of the films and droplet patterns were measured using an optical profilometer-NanoMap (1000WLI). The devices were characterised for J-V using the Keithley 2400 electronic voltmeter. The total flux and emission spectra were measured using a Konica Minolta CS-1000.

## Data availability

The data that support the findings of this study are available from the authors upon request. Source data are provided with this paper.

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

## Acknowledgements

The Visvesvaraya Ph.D. fellowship supported this research. MK acknowledges financial support from the Ministry of Electronics and Information Technology, Government of India (Grant number 2(4)/2014-PEGD (IPIIW). EBN and SCL are grateful to Australian Research Council (ARC) (DP200103036) and Department of Industry, Innovation and Science (AISRF53765) for financial support.

## Author contributions

C.K., M.K. and E.B.N. conceived the idea of the manuscript. S.K.M.M synthesised the TADF material. C.K. and A.S. performed the experiments and analysed the results with the help of S.K.M.M., M.K., S.-C.L. and E.B.N.; C.K. drafted the manuscript with the inputs of A.S. and S.K.M.M.; E.B.N., S.-C.L. and M.K. All the authors contributed to data analysis and validation as well as results discussion.

## Competing interests

The authors declare no competing interests.
