## [Peer Review File · Nature Communications]

Large area inkjet-printed OLED fabrication with solution-processed TADF inkREVIEWER COMMENTS

Reviewer #1 (Remarks to the Author):

In this manuscript, the author has developed large-area multilayer OLEDs with an inkjet-printed TADF emission layer and have successfully demonstrated that IJP as a single step maskless emission layer patterning technique to create intricate, high-resolution designs for signage, wearable electronics, and advertising without the use of typical lithography steps. However, this printing area is not very large and the strategy is not novel.

1. Why does the viscosity and surface tension decrease when two single solvents are mixed in ink solvents? Please the author provides an explanation.
2. In this article, the author prints a line from multiple points on the luminescent material, resulting in a thin film with very low surface roughness. Can the author explain in detail how to maintain the flatness of the film?
3. As is well known, temperature control can regulate the evaporation process. But the author only used three substrate temperatures of 30 °C, 40 °C and 50 °C. The author found that different substrate temperatures can lead to significant differences in patterns and processing the substrate at 40 °C can drastically reduce the coffee ring effect. However, the boiling point of the mixed ink solvent is very high, and it is difficult for smaller temperatures to have a significant impact in a short period of time. The author's experimental phenomenon contradicts this theory. Please provide a detailed explanation.
4. Why is the ink solvent ratio 6:4? Would other ratios be better?
5. In addition, please improve the language used in this article. And also please check that all references are formatted according to the specified style.

Reviewer #2 (Remarks to the Author):

The manuscript submitted to me for review is very well prepared. Despite the many articles devoted to printed emissive layers for OLED applications, there are still few examples of printed TADF systems. In my opinion, the greatest value of the work is the detailed description of the optimization of the ink and its full characterization, as well as the printing process and the evaluation of many parameters affecting the quality and thickness of the printed layers, which directly affects the performance of the devices. The authors have correctly selected the research methods, the analysis of the results presented is clear and consistent. The procedure presented in the paper can be considered as a kind of guide to optimize the process of printing thin organic layers.

In order to improve the quality of the manuscript, I recommend the authors:

-to add information regarding the significance of the modification of ACRTX synthesis used,

-to move Fig.S4 from Supporting Information to the manuscript,

-to explain whether the difference in thickness of TADF layers (Table 3) in the compared devices (e.g., SP1 ca. 36nm and IJP1 ca. 25 nm) can affect the characteristics shown in Fig.7,

-to explain in section of "OLED Device fabrication process" whether the solvent composition of the TADF ink used for printing and for the spin-cating process were the same,

-to add a more detailed analysis of the characteristics shown in Fig.7, especially the I-V and Luminance vs. Voltage curves.

In my opinion, after taking into account the above recommendations, the paper should be published in Nature Communications.

Reviewer #3 (Remarks to the Author):

This manuscript reports OLEDs with a printable TADF emissive layer. Given that there are very few reports on printable TADF OLEDs, it is quite encouraging to see this manuscript. The authors have done extensive work on the ink formulation of the TADF emissive layer. While such study of ink formulation has been reported extensively for many other emissive materials, not much has been reported for TADF inks due to solubility issues of TADF emitters.

While the work is encouraging, the current manuscript needs more information and amendments for quality, clarity and reproducibility of the work.

Here are some suggestions and questions which may help in improving the quality and clarity of the manuscript:

1. While reading the abstract and introduction, the narrative felt the study is focussed on non-chlorinated solvents, as they are more industry friendly. However, in the results, the authors started

with common chlorinated solvents and the narrative in the results section infers that the non-chlorinated solvents were chosen because the chlorinated solvents were not suitable for printing (low boiling point, etc.). Why were the chlorinated solvents studied if they were not intended to be used (which also coincidentally turned out to be not suitable)? Changing either the narrative or moving the data for chlorinated solvents to supplementary information would be more suitable.

2. Were any other non-chlorinated solvents investigated? Include reason/s why Toluene and MB were specifically chosen. Why not any other non-chlorinated solvents?

3. Why was the material ACRXTN chosen for this study?

4. Provide supporting reference for the statement for optimal surface tension "...allowing for the final surface tension to be adjusted to the optimal range (28–32 mN m⁻¹)." on page 6. OR is this a requirement for the ink-jet printer used here?

5. In Figure 2c and 2f, what are the blue squares?

6. On page 8, the authors have stated "We also discovered that reducing the number of firing nozzles from all (16) to few (5 or 1) reduced drop velocity." How many nozzles were fired for the printing of emissive layer? Are the images in Figure 4 from just one nozzle or many?

7. On page 11, the authors have stated "As shown in Figures 4c, d, & e, as the DPI was over 400, previously separate droplets began to overlap and combine to form a continuous film. Hence, we can conclude that resolution over 400 DPI are necessary to generate consistent films at 40 °C substrate temperature for our formulated ink." However, the film in 4d which corresponds to 500 DPI looks more like lines. The 3D image in supporting information doesn't add much information as the substrate and printed film have the same colour tone. Provide a cross sectional height profile similar to that of Figure S4f.

8. Looking at films in Figures 4d and 4e, film of 500 DPI looks a lot more rough as compared to film of 600 DPI. However, the AFM height data indicates the 600 DPI film is less rough (from the height scale bar). The scan area of the AFM is quite small and would not reflect what is shown in Figure 4. A cross sectional height scan over a larger area should be included. This is all the more significant, given that the OLEDs reported here are 4x4 mm².

9. Why are the emissive layer films dried under nitrogen shower? Given that the authors have argued for ambient fabrication, wouldn't it be counter intuitive to dry in a nitrogen environment. What is the effect of drying in the absence of a nitrogen shower?

10. Is the EL the same for all the OLED? Comment on that would be helpful.

11. Are the plots in Figure 7 for best performing OLEDs or for a typical representative device?

12. Statistics for OLED performance should be provided. Only data of best performing devices are included currently.

13. Scale bar for figure 4Sa and d should be increased.

RESPONSES TO REVIEWERS' COMMENTS

We are thankful to the Reviewers who provided insightful evaluations and comments. Accordingly, we have revised the manuscript in response to all the Reviewers' comments. All our modifications are shown in yellow highlighted texts while our responses to Reviewers' comments are written in blue colour font.

Reviewer #1 (Remarks to the Author):

In this manuscript, the author has developed large-area multilayer OLEDs with an inkjet-printed TADF emission layer and have successfully demonstrated that IJP as a single-step maskless emission layer patterning technique to create intricate, high-resolution designs for signage, wearable electronics, and advertising without the use of typical lithography steps. However, this **printing area** is not very large, and the strategy is not novel.

Response: We thank the Reviewer's comments. To address the Reviewer's comments on the printing areas, we have now successfully fabricated large-area IJP TADF devices with an active area of 6400 mm² (both non-patterned and patterned OLEDs). These are the largest OLED devices that we can fabricate using our facility as limited by our maximum evaporator deposition area. By using 600 DPI, we have positively demonstrated: i) large area inkjet-printed panel (LAP) OLEDs consisting of chromium gridlines on 120 × 120 mm² ITO/glass substrates with an active area of 80 × 80 mm² as shown in the new **Figure 9a**; ii) large area inkjet printed intricate design logos on 100 × 100 mm² ITO/glass substrates, namely, unpatterned and patterned large area OLEDs.

We have now added below new sections related to our new large-area IJP devices (both non-patterned and patterned OLEDs) to the manuscript (on pages 15-17):

“IJP large-area non-patterned OLEDs (with an active area of 80 × 80 mm²)

To demonstrate large area ink-jet printed (LAiP) OLEDs, we fabricated large-area panel (LAP) IJP TADF OLEDs on 120 × 120 mm² chromium/ITO/glass substrates with an active area of 80 × 80 mm² (**Figure 9a**) in a 10,000-class clean room at ambient conditions. We applied the chromium gridlines for the homogeneous current distribution as the sheet resistance of ITO starts to affect the large-area substrates. After performing an ozone surface treatment of the ITO substrate, PEDOT:PSS and PVK-TAPC layers were spin-coated on top. Substrates were then annealed at 120 °C for 15 min before the TADF ink was IJP at 600 DPI on top of the PVK-TAPC layer. This gave a device structure of ITO/PEDOT:PSS (40 nm)/PVK-TAPC (18

nm)/TADF (30 nm)/TPBi (32 nm)/Ca (20 nm)/Al (100 nm). The devices were then encapsulated for characterization and analysis.

Figure 9c shows an operational LAiP TADF OLED at 10.5 V. The device emitted homogenous light with a maximum luminescence of 622 cd m^{-2} . We noticed some non-uniformities of light emission at the lower voltages but they disappeared as the bias voltage increased. This may be due to non-uniformity of the spin-coated layers (PEDOT:PSS and PVK-TAPC) on a large substrate area. We also observed that the presence of dust particles on the substrate during the annealing process, which resulted in devices getting shorted. To minimize dust particles, substrates annealing was, therefore, performed under N_2 shower. With few iterations, we successfully fabricated and tested the large-area devices. As shown in **Figure 9c**, the LAiP OLED had the maximum current efficiency of 13.7 cd A^{-1} @ 622 cd m^{-2} , which is close to those of spin-coated small area OLEDs. The J-V-L and current efficiency plot is shown in **Supplementary Figure S11**.

Fig. 9 Images of large-area ink jet panel substrates and OLEDs. **a** Handheld large-area panel (LAiP) substrate ($120 \times 120 \text{ mm}^2$) with an active area of $80 \times 80 \text{ mm}^2$ through square chromium gridlines. **b** Inkjet patterned large area logos on a $100 \times 100 \text{ mm}^2$ substrate, consisting of *The University of Queensland* and *IIT Kanpur* with two rectangular strips of 10

$\times 70$ and 5×70 mm² before the thermal evaporation of other layers. **c** The LAP OLED (120×120 mm²) operated at 10.5 V with an inset image at 6 V. **d** The LAP (100×100 mm²) intricate an logo OLED biased at 10 V with brightness of ≈ 500 cd m⁻², measured at the rectangular strip.

J-V-L and current density of the LAP OLEDs

Fig. S11 Device characteristics of inkjet-printed large-area panel (LAP) OLEDs. a Current density and the brightness *verses* voltage of inkjet-printed large-area panel (LAP) OLED with 80×80 mm² active area. **b)** Current efficiency *verses* current density of the LAP OLED.

IJP large-area high-contrast patterned OLEDs (with an active area of 80×80 mm²)

Next, we demonstrated maskless IJP patterning technique to create intricate OLEDs on a large area substrate. The device fabrication methods are outlined in the Supplementary **Figure S12**. First, we inkjet printed a SU-8 (a negative epoxy) based photoresist dielectric (DPI 800; thickness 1 μ m) to create the desired intricate template on 100×100 mm² ITO substrates. The resulted SU-8 patterned layer was then exposed with UV light to cross-link the photoresist. HIL (PEDOT:PSS) and HTL layers (PVK-TAPC) were then spin coated on top of the SU-8/ITO template. The TADF ink was IJP at 600 DPI on the PVK-TAPC layer to create the intricate light emission regions, matching with the dielectric template. Finally, the substrates were transferred to an evaporator for thermal deposition of TPBi (32 nm) and Ca/Al electrodes. **Figure 9d** shows the proof-of-concept device consisting of logos of The University of Queensland and IIT Kanpur as well as two rectangular strips of 10×70 and 5×70 mm². The device achieved brightness of 500 cd m⁻² at 10 V. The images show high contrast glowing edges revealing intricate features. It is important to note that our negatively IJP patterned SU-8

dielectric layer eliminates direct charge recombination from TPBi to PVK-TAPC regions. This leads to OLEDs with high contrast images without the use of complicated photolithography.”

Fig. S12 Maskless IJP patterned large-area ($100 \times 100 \text{ mm}^2$) device fabrication steps. a pre-treated ITO-glass substrate. **b** inkjet printing of SU-8 (800 DPI, 1 mm thickness), a negative photoresist, to create the intricate template on the ITO substrate, followed by annealing at 80°C for 5 min and then UV exposure to crosslink the SU-8. **c** spin coating of PEDOT:PSS and PVK-TAPC layer on ITO/SU-8 substrate. **d** IJP of TADF ink on top of the PVK-TAPC layer to create light emission regions. **e** thermal deposition of ETL TPBi, and Ca as well as an aluminium electrode.

1. Why does the viscosity and surface tension decrease when two single solvents are mixed in ink solvents? Please, the author provides an explanation.

Response: We thank the Reviewer’s valid question. The viscosity and surface tension of an ink solution made from two separate solvents may differ from those of the individual solvents. This happens because of the intermolecular interaction of the two solvents, which might modify their physical characteristics. Both solvents utilized in this study have distinct viscosity and surface tension values. Namely, toluene has a viscosity of 0.8 cP, whereas MB has a viscosity value of 2.0 cP. When toluene and MB are combined in a ratio of 60:40, the solution's viscosity was determined to be 1.3 cP. The addition of 50 mg of material to the solution was not significant to impact the global solution viscosity.

Furthermore, the surface tension at 20 °C for toluene and MB are 28 mN and 38 mN m⁻¹, respectively. As a binary solvent system, the surface tension falls between these two values. The most commonly used model to predict the viscosity of a binary solvent mixture is the empirical equation known as the Grunberg-Nissan equation¹:

$$\eta_{\text{mix}} = \sum \chi_i \cdot \eta_i,$$

where η_{mix} is the viscosity of the mixture, η_i is the viscosity of each pure solvent, and χ_i is the volume fraction of each solvent in the mixture. The type of the solvents, the interactions between their molecules, and the concentration of the solutes are some examples of the factors that affect the size and direction of the change in surface tension that occurs during mixing. Recently, Kleinheins *et al.* have compiled an overview of the most popular models and their ability to reproduce experimental data of ten binary aqueous solutions². It is important to note that the specific behaviour of solvent mixing and its impact on viscosity and surface tension can vary depending on the solvents and their concentrations. Experimental measurements and theoretical models are typically used to understand and predict these changes^{3,4}.

2. In this article, the author prints a line from multiple points on the luminescent material, resulting in a thin film with very low surface roughness. Can the author explain in detail how to maintain the flatness of the film?

Response: We are grateful for the Reviewer's remarks. First, we printed at lower resolution (*i.e.*, lower DPI) so that the individual drop diameter under the printing conditions can be measured. In this study, at 200 DPI there was clear separation of each droplet, and a centre-to-centre (C2C) distance of 125 μm was measured (Figure R1a). The droplet diameter slightly varied with different printing parameters. Then we input these values into a droplet simulator to determine the DPI, where the droplets were overlapped. Insufficient droplet overlap (Figure R1b, c) led to non-uniform films and high surface roughness, while too much droplet overlaps wasted ink and increased drying time, leading to heterogeneous films. We have found that under the study conditions, >400 DPI, were the ideal printing resolution for uniform films with complete merger of droplets (Figure R1d) with a low surface roughness.

Fig. R1 Simulation of TADF films for homogeneous film formation with 200, 300, and 400 and 500 DPI, respectively. **a** When DPI is 200, the centre-to-centre (C2C) distance is 125 μm . **b** only further increasing the DPI to 300 reduces the distance to 83.3 μm between the printed droplets. **c** Droplets starts to merge as C2C distance (62.5 μm) is less than the drop diameter (75 μm). **d** at 500 DPI C2C distance reduces to 50 μm and droplets merge with completely overlapped area.

3. As is well known, temperature control can regulate the evaporation process. But the author only used three substrate temperatures of 30, 40, and 50 $^{\circ}\text{C}$. The author found that different substrate temperatures can lead to significant differences in patterns and processing the substrate at 40 $^{\circ}\text{C}$ can drastically reduce the coffee ring effect. However, the boiling point of the mixed ink solvent is very high, and it is difficult for smaller temperatures to have a significant impact quickly. The author's experimental phenomenon contradicts this theory. Please provide a detailed explanation.

Response: We thank the Reviewer's comments. The droplets had a minimal volume of 10 μL (*i.e.*, one millionth of a microlitre), and after ejecting from the nozzle, they were impinging on the substrate, which had an average diameter of 50-70 μm and a thickness of a few tens of nanometres. The droplets' thickness in nm and diameter in μm scale make them a large surface-

to-volume ratio, meaning the droplets will be increasingly affected by slight variations in substrate temperature as well as the solvent vapour pressure.

The coffee ring effect originated from the capillary flow induced by the differential evaporation rates between the edge and interior of the TADF ink drop, directly affecting the uniformity of printed TADF films.⁵ To obtain uniform TADF films, the printing parameters require systematic optimization. Based on the formation mechanism of the coffee ring effect, there are three methods for inhibiting it: (i) weakening the capillary flow from the inside to the outside of the droplet, (ii) increasing the Marangoni flow from the outside inside the liquid, and (iii) controlling the movement of the three-phase contact line of liquid drops in the drying process.

To balance the capillary flow and eliminate the coffee ring effect, Sun *et al.* optimized the evaporation rate of their CsPbBr₃ perovskite QD ink by altering the volume of two solvents (dodecane and toluene) to form appropriate Marangoni flow.⁶ Similarly to our case, at a lower substrate temperature of 30 °C, the capillary forces dominated, and the solvent did not evaporate sufficiently due to the high boiling point and low vapor pressure of MB. As we increased the substrate temperature to 40 °C, there was an optimal equilibrium between the Marangoni and viscous capillary forces, suppressing the coffee ring effect. As we increased the substrate temperature further to 50 °C, we obtained rougher films due to quick evaporation of the solvent post deposition. Namely, when the substrate temperature is too high, the droplets cannot merge in liquid form, and instead becomes a liquid ink addition to an already solid film. We note that Amrut *et al.*⁷ made similar observation of substrate temperature on final film quality.

4. Why is the ink solvent ratio 6:4? Would other ratios be better?

Response: We are thankful for the Reviewer's valid questions. We have achieved closer values concerning a printable range of surface tension of (28-32 mN m⁻¹) for DMC Samba cartridges for inkjet printing specified by the company. Further increasing the MB ratio in the formulated ink leads to surface tension values of 35.4 mN m⁻¹ (Figure R2) outside this ideal printable window.

Fig. R2 Surface tension value of 50:50 (Toluene:MB) to be 35.4 mN m^{-1} .

5. In addition, please improve the language used in this article. And also, please check that all references are formatted according to the specified style.

Response: We thank the Reviewer's remarks. We have carefully proofread the MS and SI, modified our language used in this article, and re-formatted some of our references to fully in line with the journal style. These updates can be found in yellow highlights.

Reviewer #2 (Remarks to the Author):

The manuscript submitted to me for review is very well prepared. Despite the many articles devoted to printed emissive layers for OLED applications, there are still few examples of printed TADF systems. In my opinion, the greatest value of the work is the detailed description of the optimization of the ink and its full characterization, as well as the printing process and the evaluation of many parameters affecting the quality and thickness of the printed layers, which directly affects the performance of the devices. The authors have correctly selected the research methods, the analysis of the results presented is clear and consistent. The procedure presented in the paper can be considered as a kind of guide to optimize the process of printing thin organic layers.

1. In my opinion, after taking into account the above recommendations, the paper should be published in Nature Communications.

Response: We thank the Reviewer for valuable suggestions and recommendations for the publication.

2. In order to improve the quality of the manuscript, I recommend the authors to add information regarding the significance of the modification of ACRTX synthesis used.

Response: We are grateful for the Reviewer's comment. The synthesis of ACRXTN is essentially the same as that described in the literature (Ref S1), including the reaction conditions, work-up and purification processes as well as the reaction yields. The exception being the ligand used in this study was PCy₃HBF₄, which is much cheaper (\$47.9/g sourced from Sigma-Aldrich, <https://www.sigmaaldrich.com/AU/en>) than that (A\$259.0/g) of P(t-Bu)₃HBF₄ reported in Ref S1.

3. to move Fig.S4 from Supporting Information to the manuscript,

Response: We thank the Reviewer for the helpful suggestion. We have now moved the 3D image of printed lines of Figure S4 from the SI to the manuscript as the new Figure 4g.

Fig. 4 The microscopic image of printed droplets in $4 \times 4 \text{ mm}^2$ square patterns with different DPIs on substrates at $40 \text{ }^\circ\text{C}$. a Printed with 200 DPI. b 300 DPI. c 400 DPI. d 500 DPI. e 600 DPI. f Printed lines with 200 to 800 DPI from L1 to L7, respectively. g 3D-images of IJP single lines L1 to L7 (200 to 800 DPI).

4. Explain whether the difference in thickness of TADF layers (Table 3) in the compared devices (e.g., SP1 ca. 36nm and IJP1 ca. 25 nm) can affect the characteristics shown in Fig.7,

Response: We agree with the Reviewer that these variations in the emissive layer thickness can impact the device performance. We observed that higher thickness increased the turn-on voltages and reduces the device efficiency.

We have now added the following text to the main texts (page 12);

“We note that higher thickness increased the light turn-on voltages and reduced the device efficiency.”

5. Explain in section of "OLED Device fabrication process" whether the solvent composition of the TADF ink used for printing and for the spin-coating process were the same,

Response: We are grateful for Reviewer’s comment. Yes, the solvent composition for inkjet-printed and spin-coated devices are the same.

We have now added the missing statement to the main (page 12) as:

“The solvent composition for inkjet-printed and spin-coated devices are the same...”

6. add a more detailed analysis of the characteristics shown in Fig.7, especially the I-V and Luminance vs. Voltage curves.

Response: We thank the Reviewer's comment. We have now added more detailed analysis of the device characteristics to the main texts on page 12 (with yellow highlights).

“We note that higher thickness increases the turn-on voltages and reduces the device efficiency. **Table 3** summarizes device characteristics of 3 to 4 pixels per each device. At 1000 cd m^{-2} , we observed that pixel to pixel variation is very less ($\sim 5\%$), indicating uniform and reproducibility of IJP films. The maximum external quantum efficiency (EQE) was $\sim 4.7\%$ @ 30 cd m^{-2} for spin-coated and $\sim 4.3\%$ @ 30 cd m^{-2} for the IJP OLEDs (**Figure S9**). It is interesting to note that the efficiency roll-off of the IJP devices were comparable to the spin-coated device (**Figure S10**). At a higher brightness level, the efficiency of IJP OLEDs and spin coated OLEDs were $\approx 7 \pm 1 \text{ cd A}^{-1}$ @ 1000 cd m^{-2} . This suggests, charge balance is maintained in the simple device structure (without blocking layers) and exciton annihilation is negligible in the emissive layer. The EL spectra of the spin coated (SP1) and IJP1 (500 DPI), IJP2 (600 DPI) & IJP3 (700 DPI) at 1000 cd m^{-2} were almost identical (Figure 7d). This suggest, exciton formation and recombination photo physics of inkjet-printed OLEDs were identical to the spin coated OLEDs”

Fig. S9 External quantum efficiency *versus* luminance of the inkjet-printed OLEDs with DPI of 500, 600 and 700 for IJP1, 2 and 3, respectively) and spin-coated devices (SP1).

Reviewer #3 (Remarks to the Author):

This manuscript reports OLEDs with a printable TADF emissive layer. Given that there are very few reports on printable TADF OLEDs, it is quite encouraging to see this manuscript. The authors have done extensive work on the ink formulation of the TADF emissive layer. While such a study of ink formulation has been reported extensively for many other emissive materials, not much has been reported for TADF inks due to the solubility issues of TADF emitters.

While the work is encouraging, the current manuscript needs more information and amendments for the quality, clarity, and reproducibility of the work. Here are some suggestions and questions which may help in improving the quality and clarity of the manuscript.

Response: We thank the Reviewer for the valuable suggestions and recommendations of the publication.

1. While reading the abstract and introduction, the narrative felt the study is focused on non-chlorinated solvents, as they are more industry-friendly. However, in the results, the authors started with common chlorinated solvents, and the narrative in the results section infers that the non-chlorinated solvents were chosen because the chlorinated solvents were not suitable for printing (low boiling point, etc.). Why were the chlorinated solvents studied if they were not intended to be used (which also coincidentally turned out to be not suitable)? Changing either the narrative or moving the data for chlorinated solvents to supplementary information would be more suitable.

Response: Thank you for the helpful comments and suggestion. We have now moved the chlorinated solvents data and updated discussion to **Supplementary information 2** and added the following paragraph to the main texts on page 5 (with yellow highlights).

“To create a homogeneous and smooth emissive thin film, the first step was to select suitable solvents and concentrations for the emissive blend containing ACRXTN TADF guest and CBP host. The ink formulation using combination of non-chlorinated solvents with a low and a high boiling point was tried, where more details with other solvents are given in the Supplementary Section 2 and Table S1. We found that while low boiling point solvents evaporate rapidly at the air–nozzle interface, a higher boiling point solvent (*e.g.*, *o*-DCB) solves the evaporation issue but it restricts the drying mechanism of drops after jetting, and chlorinated solvents lead to poor long term ink stability. Hence, we used non-chlorinated solvents of toluene and methyl benzoate (MB) in this work as a binary solvent system”.

2. Were any other non-chlorinated solvents investigated? Include reason/s why Toluene and MB were specifically chosen. Why not any other non-chlorinated solvents?

Response: We are grateful for the Reviewer's comments. Apart from toluene and MB, we haven't employed other non-chlorinated solvents. To fall within the ideal printing window using Dimatix cartridges in this work, we need an ink formulation with surface tension range of 28–32 mN m⁻¹ and viscosity range of 1–10 cP. The ink should not evaporate from the nozzles when printer is in non-jetting mode, but should have rapidly evaporate after imping on the substrate to avoid the coffee ring effect. It is important to note that a low boiling point (bp) solvent like toluene (bp = 110 °C) with a high vapor pressure (0.13 atm at 25°C) rapidly evaporates after drop impinging on the substrate, whereas a high boiling point solvent like MB (bp ≈ 200 °C) with a low vapour pressure (50×10^{-5} atm at 25 °C) helps to prevent clogging of the nozzle, especially in non-jetting mode. Hence, our use of them as a binary system is beneficial to have viscosity (toluene = 0.8 cP and MB = 2.0 cP) and surface tension (toluene = 28 mN m⁻¹ and MB = 37.6 mN m⁻¹) in the range of printable windows.

Please refer to the images in Figure S2b while using single low boiling point solvents.

3. Why was the material ACRXTN chosen for this study?

Response: We thank the Reviewer's query. ACRXTN was chosen as the TADF emitter used in the work mainly because of its reported high RISC rate ($>10^6$ s⁻¹), high singlet radiative rate (with a prompt decay lifetime of ≈20 ns) and high film PLQYs (≈90%) in both neat films as well as host-guest blend. These advantageous properties lead to comparatively superior device performance with high brightness and relatively low EQE roll-off. Further, compared to multi-step chemical synthesis of other TADF materials (*e.g.*, CzIPN derivatives), ACRXTN has single donor-acceptor moieties, which allows for the use of simple and inexpensive synthetic procedures from readily available precursors as shown in the Supplementary information.

4. Provide a supporting reference for the statement for optimal surface tension "...allowing for the final surface tension to be adjusted to the optimal range (28–32 mN m⁻¹)." on page 6. OR is this a requirement for the inkjet printer used here?

Response: We are grateful for the Reviewer's comment. We used Fujifilm Dimatix Samba Cartridges (DMC-11610) in this study. The company has a datasheet of the cartridge, specifying the ideal fluid requirement ranges for surface tension values of 28–32 mN m⁻¹, see <https://asset.fujifilm.com/www/us/files/2021->

[04/ae8a1e167ce8c273fcdd31ecffd9ec80/PDS00142.pdf](https://doi.org/10.1039/C4/ae8a1e167ce8c273fcdd31ecffd9ec80/PDS00142.pdf) for more specifications. Other researchers have verified the values⁷⁻⁹.

We have added the related statement in the texts (on page 5);

“(28–32 mN m⁻¹) of the Fujifilm Dimatix Samba Cartridges (DMC-11610) used in this study”

5. In Figures 2c and 2f, what are the blue squares?

Response: We apology for the omission. These blue squares are called "*Dropview camera screen*" tabs to capture the drop velocity and drop volume, respectively, corresponding to firing nozzle as shown in Figure R3 where a total of 5 nozzles are used for drop ejection, and nozzle no. 3 is in the center.

Fig. R3 Dropview camera screen tabs of LP50-Pixdro inkjet printer, showing 5 nozzles with nozzle no. 3 in the center.

6. On page 8, the authors have stated "We also discovered that reducing the number of firing nozzles from all (16) to few (5 or 1) reduced drop velocity." How many nozzles were fired for the printing of the emissive layer? Are the images in Figure 4 from just one nozzle or many?

Response: We thank the Reviewer for the comment. We used 5 nozzles for inkjet printing the emissive layer as shown in Figure 4. We observed that when we used all 16 nozzles, the drop velocity was higher than that of 5 nozzles. We have mentioned this phenomenon on page 7, so that the same drop velocity and volume values could be achieved by repeating the experiment with the same drop ejection parameters, such as pulse width and applied pulse voltages. In the

literature, we found that Cao *et al.* also noted that drop velocity changes due to multi-nozzle interference¹⁰.

7. On page 11, the authors have stated "As shown in Figures 4c, d, & e, as the DPI was over 400, previously separate droplets began to overlap and combine to form a continuous film. Hence, we can conclude that resolution over 400 DPI are necessary to generate consistent films at 40 °C substrate temperature for our formulated ink." However, the film in 4d which corresponds to 500 DPI looks more like lines. The 3D image in supporting information doesn't add much information as the substrate and printed film have the same colour tone. Provide a cross-sectional height profile similar to that of Figure S4f.

Response: The Reviewer is correct; the specific image has some non-uniformity owing to the misdirected of the nozzles. Nevertheless, we have updated the image with another printed pattern deposited under the identical conditions (500 DPI) (as shown in the updated Figure 4d). (We have also added the 3D image to **Figure S5** and the cross-section height profiles of printed patterns to **Figure S5f**—see below Response.)

Fig. 4 The microscopic image of printed droplets in $4 \times 4 \text{ mm}^2$ square patterns with different DPIs on substrates at 40 °C. **a** Printed with 200 DPI. **b** 300 DPI. **c** 400 DPI. **d** 500 DPI. **e** 600 DPI. **f** Printed lines with 200 to 800 DPI from L1 to L7, respectively. **g** 3D-images of IJP single lines L1 to L7 (200 to 800 DPI).

Fig. S5 3D images and cross sectional height profiles. The 3D images of printed droplets patterns with different DPIs at 40 °C substrate temperature, **a** Printed with 200 DPI, **b** 300 DPI, **c** 400 DPI, **d** 500 DPI, **e** and 600 DPI. **f** Cross sectional height profiles.

8. Looking at films in Figures 4d and 4e, film of 500 DPI looks a lot more rough as compared to film of 600 DPI. However, the AFM height data indicates the 600 DPI film is less rough (from the height scale bar). The scan area of the AFM is quite small and would not reflect what is shown in Figure 4. A cross sectional height scan over a larger area should be included. This is all the more significant, given that the OLEDs reported here are $4 \times 4 \text{ mm}^2$.

Response: We thank the Reviewer for the comment. On a wider scale (mm), the film is showing waviness; however, at the micro level films are smooth. We have added the cross-section profile image of 500 to 700 DPI films to Supplementary Information as Figure S5f.

(f) Height Profile

Fig. S5f Height profiles of 500, 600 and 700 DPI.

9. Why are the emissive layer films dried under a nitrogen shower? **Given that the authors have argued for ambient fabrication**, wouldn't it be counterintuitive to dry in a nitrogen environment. What is the effect of drying in the absence of a nitrogen shower?

Response: We thank the Reviewer's comment. Drying in a nitrogen shower would in principle benefit device performance although we have not observed much difference between with and without N₂ shower. However, a problem of dust particles landed on the films was encountered for large area devices fabrication while annealed on a hotplate without a N₂ shower. We, therefore, opted to use a metal cover to protect the devices from dust particles. However, this prevented the solvent from sufficiently evaporating. As a result, we used a N₂ shower as a carrier gas to assist the volatile solvents to the outlet (**Figure R4**). The flow rate of the nitrogen gas was kept low to not disrupt the film surface. This method has provided a route to reduce the micro-particles and to protect large-area devices during annealing.

Fig. R4 The N₂ shower used in our annealing the inkjet-printed films over a hotplate under.

10. Is the EL the same for all the OLEDs? Commenting on that would be helpful.

Response: We thank the Reviewer's remark. As shown in the figure below, the EL spectra of the inkjet-printed and spin-coated devices are essentially the same (with peaks at 502 and 500 nm, respectively). We have now added the inkjet-printed and spin-coated EL spectra to **Figure 7d**. The updated statement is as following on page 13;

“The essentially identical EL spectra of the spin coated (SP1) and IJP1 (500 DPI), IJP2 (600 DPI) & IJP3 (700 DPI) at 1000 cd m⁻² can be observed as shown in Figure 7d with an EL peak at ≈502 nm.”

Fig. 7d. EL spectra of spin coated (SP1) and the inkjet-printed TADF OLED devices (IJP1, IJP2, and IJP3).

11. Are the plots in Figure 7 for best-performing OLEDs or for a typical representative device?

Response: We are thankful for the Reviewer’s comment. Our standard device comprises 8 pixels per device, and we have characterized 3 to 4 pixels for each device. Figure 7 have plotted the average data of these devices.

12. Statistics for OLED performance should be provided. Only data of best-performing devices are included currently.

Response: We thank the Reviewer’s comment. We have now updated **Table 3** in the manuscript by accompanying the device statistics image in the Supplementary information (**Figure S10**) to reflect the addition of all 3 to 4-pixel data variation for each device (SP1, IJP1, IJP2, and IJP3) with the inclusion of error bars for each device.

Table 3: Summary of V_{on} , LE_{max} , L , and CIE coordinates of the fabricated devices. Device pixel area $4 \times 4 \text{ mm}^2$.

Devices	TADF Thickness (nm)	V_{on} (V)	$CE (cd A^{-1})$ @ 100 cd m^{-2}	$CE (cd A^{-1})$ @ 1000 cd m^{-2}	L_{max} ($cd m^{-2}$)
SP1	36 ± 2	3.9 ± 0.1	13.7 ± 0.8	7.6 ± 0.8	1775 ± 80
IJP1 (DPI=500)	25 ± 2	4.1 ± 0.1	12.4 ± 1.0	6.8 ± 0.5	2350 ± 40
IJP2 (DPI=600)	29 ± 2	4.2 ± 0.1	7.5 ± 2.7	6.0 ± 0.2	2060 ± 90
IJP3 (DPI=700)	33 ± 3	4.3 ± 0.1	9.5 ± 0.4	6.9 ± 0.6	1985 ± 10

Also, we have added the following new **Figure S10** to the Supplementary information.

Fig. S10 Performance statistics of small area OLEDs. The device performance metrics were obtained from 3 to 4 pixels on one substrate. **a** Light turn-on voltage variation of spin coated and IJP devices measured at $\approx 1 \text{ cd m}^{-2}$. **b** Brightness from IJP and spin coated OLEDs. **c** Current

efficiency performance of IJP and spin coated OLEDs. The maximum current efficiency CE_{\max} is quoted for a luminance of greater or equal to $\approx 10 \text{ cd m}^{-2}$. CE_{100} and CE_{1000} is the current efficiency at 100 cd m^{-2} and 1000 cd m^{-2} , respectively.

13. Scale bar for Figure 4Sa and d should be increased.

Response: We thank the Reviewer's helpful comment. We have now made the required update to the Supplementary Information as shown below.

Fig. S4 **a** 2D-image of the printed droplets with 200 DPI. **d** 2D image of the printed droplets at 600 DPI but after drying the film showing ununiform as many patches can be seen.

References

1. Grunberg, L. & Nissan, A. H. Mixture law for viscosity. *Nature* **164**, 799–800 (1949).
2. Kleinheins, J. *et al.* Surface tension models for binary aqueous solutions: a review and intercomparison. *Phys. Chem. Chem. Phys.* **25**, 11055–11074 (2023).
3. Maher, J. Y. Binary Liquid Gels. in Boccara, N., Daoud, M. (eds) *Physics of Finely Divided Matter*. Springer Proceedings in Physics, vol 5. Springer, Berlin, Heidelberg. **70**, 356–360 (1985).
4. Shereshefsky, J. L. A theory of surface tension of binary solutions. I. Binary liquid mixtures of organic compounds. *J. Colloid Interface Sci.* **24**, 317–322 (1967).

5. Tan, H. *et al.* Evaporation-triggered microdroplet nucleation and the four life phases of an evaporating Ouzo drop. *PNAS* **113**, 8642–8647 (2016).
6. Gao, A. *et al.* Printable CsPbBr₃ perovskite quantum dot ink for coffee ring-free fluorescent microarrays using inkjet printing. *Nanoscale* **12**, 2569–2577 (2020).
7. Amruth, C. *et al.* Inkjet printing of thermally activated delayed fluorescence (TADF) dendrimer for OLEDs applications. *Org. Electron.* **74**, 218–227 (2019).
8. De Gans, B. J. & Schubert, U. S. Inkjet printing of polymer micro-arrays and libraries: Instrumentation, requirements, and perspectives. *Macromol. Rapid Commun.* **24**, 659–666 (2003).
9. Hoeng, F., Bras, J., Gicquel, E., Krosnicki, G. & Denneulin, A. Inkjet printing of nanocellulose-silver ink onto nanocellulose coated cardboard. *RSC Adv.* **7**, 15372–15381 (2017).
10. Ye, Y. *et al.* Numerical analysis of droplets from multinozzle inkjet printing. *J. Phys. Chem. Lett.* **11**, 8442–8450 (2020).

REVIEWERS' COMMENTS

Reviewer #1 (Remarks to the Author):

The authors have corrected the manuscript accordingly. I have no further comments.

Reviewer #2 (Remarks to the Author):

I would like to thank the authors of the publication for the explanations they have prepared and for taking into account my comments in the manuscript, mainly concerning the addition of a more detailed analysis of the characteristics of the OLEDs tested (Fig.7). I stand by my first review that the paper should be published in Nature Communications, as the manuscript contains a well-described procedure for the optimisation of the printing process of emissive organic films, starting from the preparation and full characterisation of the ink to the characterisation of the devices obtained. It was a pleasure for me to review this manuscript.

Reviewer #3 (Remarks to the Author):

The authors have addressed all my concerns and comments satisfactorily. They have made significant changes to the manuscript and improved the quality and reproducibility.